# Friction Stir Welding of AA5754-H24: Impact of Tool Pin Eccentricity and Welding Speed on Grain Structure, Crystallographic Texture, and Mechanical Properties

**DOI:** 10.3390/ma16052031

**Published:** 2023-03-01

**Authors:** Mohamed M. Z. Ahmed, Ahmed R. S. Essa, Sabbah Ataya, Mohamed M. El-Sayed Seleman, Ali Abd El-Aty, Bandar Alzahrani, Kamel Touileb, Ashraf Bakkar, Joffin J. Ponnore, Abdelkarim Y. A. Mohamed

**Affiliations:** 1Department of Mechanical Engineering, College of Engineering at Al Kharj, Prince Sattam bin Abdulaziz University, Al Kharj 11942, Saudi Arabia; 2Department of Metallurgical and Materials Engineering, Faculty of Petroleum and Mining Engineering, Suez University, Suez 43512, Egypt; 3Mechanical Department, Faculty of Technology and Education, Suez University, Suez 43512, Egypt; 4Faculty of Engineering, King Salman International University, El-Tor 45615, Egypt; 5Department of Mechanical Engineering, College of Engineering, Imam Mohammad Ibn Saud Islamic University, Riyadh 11432, Saudi Arabia; 6Mechanical Engineering Department, Faculty of Engineering-Helwan, Helwan University, Cairo 11795, Egypt; 7Department of Environmental Engineering, College of Engineering at Al-Leith, Um Al-Qura University, Al-Lith 28434, Saudi Arabia

**Keywords:** friction stir welding, AA5754-H24 aluminum alloy, tool pin eccentricity, welding speeds, microstructure, crystallographic texture, mechanical properties

## Abstract

This study investigates the effect of tool pin eccentricity and welding speed on the grain structure, crystallographic texture, and mechanical properties of friction stir welded (FSWed) AA5754-H24. Three tool pin eccentricities of 0, 0.2, and 0.8 mm at different welding speeds ranging from 100 mm/min to 500 mm/min and a constant tool rotation rate of 600 rpm were investigated. High-resolution electron backscattering diffraction (EBSD) data were acquired from each weld’s center of the nugget zone (NG) and processed to analyze the grain structure and texture. In terms of mechanical properties, both hardness and tensile properties were investigated. The grain structure in the NG of the joints produced at 100 mm/min, 600 rpm, and different tool pin eccentricities showed significant grain refining due to dynamic recrystallization with average grain sizes of 18, 15, and 18 µm at 0, 0.2, and 0.8 mm pin eccentricities, respectively. Increasing the welding speed from 100 to 500 mm/min further reduced the average grain size of the NG zone to 12.4, 10, and 11 µm at 0, 0.2, and 0.8 mm eccentricity, respectively. The simple shear texture dominates the crystallographic texture with both $\bar{B}$/B texture component with the C component at their ideal positions after rotating the data to align the shear reference frame with the FSW reference frame in both the PFs and ODF sections. The tensile properties of the welded joints were slightly lower than the base material due to the hardness reduction in the weld zone. However, the ultimate tensile strength and the yield stress for all welded joints increased by increasing the friction stir welding (FSW) speed from 100 to 500 mm/min. Welding using the pin eccentricity of 0.2 mm resulted in the highest tensile strength; at a welding speed of 500 mm/min, it reached 97% of the base material strength. The hardness profile showed the typical W shape with a reduction in the hardness of the weld zone and a slight recovery of the hardness in the NG zone.

## 1. Introduction

Friction stir welding (FSW) is an innovative solid-state welding technology in which the material flow around the FSW tool plays a vital role in forming the joint between the two plates or sheets [1,2,3,4,5]. Understanding the material flow behavior around the tool will help significantly achieve high-quality joints. Thus, since the FSW invention by Wyanne Thomas at the TWI in 1991 [6], numerous studies have focused on the material flow pattern [1,7,8,9,10,11,12,13,14,15]. Ying et al. [8,10] investigated the flow patterns during the FSW of AA2024 to AA6061 and observed that the flow was found to be complex spirals and vortex-like and affected by the tool rotation rate. Guerra et al. [9] studied the flow pattern during FSW using a faying surface tracer and a nib frozen in place. They reported a significant vertical or vortex movement of material within the rotational zone caused by the tool’s profile due to extreme deformation. Kumar and Kailas [12] investigated the role of the FSW tool in material flow and joint formation. They reported that the friction stir weld formation involves two different material flow regimes: “pin-driven flow” and “shoulder-driven flow”. These materials’ flow regimes merge to form a defect-free weld. These two different modes will be more distinguished and affect the welded joints’ various aspects as the welded plates’ thickness increases. Ahmed et al. [16] investigated the effect of tool geometry on the grain structure and texture of the thick section AA6082. They studied the effect of parallel and tapered pins on the joints of 32 mm thickness and found that the use of a tapered probe tool results in a significant variation in the grain size from the top to the base of the NG, whereas parallel probe tools produce a uniform grain size throughout the NG [17]. Elangovan et al. [18] studied the influences of tool pin profile and tool shoulder diameter on the formation of friction stir processing zone and reported that the pin profile has a crucial role in material flow. It regulates the welding speed of the FSW process. The pin is generally cylindrical, frustum tapered, threaded, or flat. Pin profiles with flat faces (square or triangular) are sometimes associated with eccentricity, which allows incompressible material to pass around the pin profile. One of the tool aspects that impacts the properties and microstructure of the weld nugget is the offsite of the tool pin axis relative to the shoulder axis, known as tool pin eccentricity. Thus, Thomas and Nicholas [19] reported that tool pin eccentricity is associated with a dynamic orbit, which becomes a part of the FSW process. Also, the effect of tool eccentricity during FSW was studied by several researchers [1,4,5,20]. The effect of tool pin eccentricity on the microstructure and mechanical properties of the thick plate friction stir welded (FSWed) 7075 aluminum alloy has been investigated using a tapered threaded pin [21]. Shah et al. [4] studied the effect of pin eccentricity on the properties of 6061 aluminum alloy welded joints. They noticed that the NG zone expands, which indicates the enhancement of the material flow of the NG area due to using an eccentric tool. Moreover, they concluded that using an eccentric tool with 0.2 mm eccentricity increases the soft region of the NG but has no pronounced effect on the mechanical properties of the 6061 aluminum alloy. Also, Chen et al. [22] studied the effect of tool pin eccentricity on the microstructure and mechanical properties of friction stir processed Al-5052 alloy. They applied different tool eccentricities. They indicated that the stir zone produced by tool pin eccentricity of 0.4 mm showed the highest hardness and yield strength and attributed that to the enhanced grain-boundary strengthening. Recently, Su et al. [23] investigated the effect of tool eccentricity on material flow during FSW using numerical simulation based on computational fluid dynamics. They reported that tool eccentricity in the FSW process causes the transformation of material flow velocity around the tool, especially the evident variation of material flow direction, which starts formation of the periodic feature. Also, Shah et al. [24] investigated the effect of tool eccentricity on the strain rate and microtexture of rapid-quenched AA6061 friction stir welds. They reported that stir zone grain refinement is observed in all samples, with the eccentric sample having the finest grains (3.18 μm). In terms of macrotexture, they reported that the aligned samples have a strong {11-2}<1-10> B simple shear texture component, while the eccentric sample shows morphological changes towards a strong {11-1}<1-10> A simple shear texture component.

Finally, based on the above literature, the material flow behavior during the stirring process has been regarded as one of the most crucial aspects of FSW and/or FSP, which may be directly related to tool eccentricity rather than to tool geometry as previously thought. Accordingly, it can be stated that tool eccentricity is a key parameter during the FSW process and has a significant impact on both microstructure and crystallographic texture. In addition, this area of research still needs more studies on different alloy materials for a full understanding. Thus, the current study investigates the effect of tool pin eccentricity on the microstructure, crystallographic texture, and mechanical properties of AA5754 aluminum alloy. More specifically, the work focuses on studying the effect of tool pin eccentricity in the deviation of the crystallographic texture components as an indication for the complex material flow occurring during FSW. Electron backscattering diffraction (EBSD) will be used to investigate grain structure and crystallographic texture.

## 2. Experimental Procedure

### 2.1. Friction Stir Welding

Friction stir welds of 5754-H24 aluminum alloy in butt joints were performed using a homemade FSW machine [25,26,27]. The chemical composition and the mechanical properties of as-received AA5754-H24 are listed in Table 1 and Table 2, respectively. The welded plates’ dimensions were 5 mm thick, 100 mm wide, and 120 mm long.

The FSW tools used for the welding process were machined from 40 mm-diameter rod of W302 cold worked tool steel (0.39% C, 5.2% Cr, 0.1% Si, 0.95% V, 0.40% Mn, 1.4% Mo, and 91.56 wt% Fe), and heat-treated to attain 62 HRC. Three FSW tools of cylindrical pins were designed and machined to achieve different eccentricities. The first tool was without pin eccentricity (e = 0). The second was with an eccentricity of 0.2 mm (e = 0.2 mm), where the pin axis was shifted by 0.2 mm from the shoulder axis. The last tool was with an eccentricity of 0.8 mm (e = 0.8 mm). For all the tools, the shoulder had a diameter of 19 mm and 2° concavity in the inner surface. In addition, the pin dimensions were kept constant at 4.6 length and 6 mm diameter. The schematic drawing of the three FSW tools with the different pin eccentricities are given in Figure 1. The FSW experiments of the AA5754-H24 aluminum butt joints were carried out at different welding speeds of 100, 200, 300, and 500 mm/min. The other FSW parameters related to the tool rotation rate, tool plunge depth, and tilt angle were kept constant at 600 rpm, 0.2 mm, and 3°, respectively.

### 2.2. Evaluation of the Welded Joints

The FSWed joints were evaluated via visual inspection, macro-examination, microstructure, tensile properties, and joint efficiency. The macroscopic examination of cross-sections perpendicular to the welding direction (WD) of the FSWed joints were carried out after the standard metallographic preparation up to 0.05 alumina polishing surface and followed by etching with solution contains 10 g NaOH dissolved in 90 mL distilled water to reveal the macro features. For the EBSD investigation, selected samples from the weld NG of the produced joints were cut and prepared to assess the grain structure and texture. The preparation process was to subject these samples to mechanical polishing and electropolishing via a solution of 30% nitric acid (HNO_3_) in methanol (CH_3_OH) for 60 s at 14 V and −15 °C. A FEI Quanta FEG 250 Field Emission Gun Scanning Electron Microscope (FEGSEM), FEI Company (Hillsboro, OR, USA), equipped with a Hikari EBSD camera controlled by EDAX-OIM7 analysis software, was used for EBSD data acquisition and post-processing.

Tensile testing was performed using a 300 KN capacity universal tensile testing machine (Instron 4210, Norwood, MA, USA) at a strain rate of $0.001\;s^{-1}$. The tensile specimens were cut perpendicular to the WD, with the sample dimensions depicted in Figure 2. Vickers hardness testing machine (HWDV-75, TTS Unlimited, Osaka, Japan) was used to evaluate the hardness profiles along the width of the weld samples using a load of 1.0 kg and dwell time of 15 s.

## 3. Results and Discussion

### 3.1. Joint Top View and Macro Examination of Transverse Cross-Sections

Figure 3 illustrates the top view of the AA5754-H24 FSWed joints produced at a constant rotation rate of 600 rpm and different welding speeds from 100 to 500 mm/min using the FSW tools with different pin eccentricities of 0, 0.2, and 0.8 mm. The top surface of the welded joints is mainly influenced by the shoulder design and its surface finish. It can be seen that the top surface appearance of the produced 5754-H24 joints is fairly similar except that FSWed at the highest welding speed of 500 mm/min, especially at eccentricities of 0.2 and 0.8 mm there is a little flash on the advancing side (AS) and the retreating side (RS) has been noted. In terms of welding speed, it is clear that the appearance of the top surface of welded joints is affected by the welding speed. The semicircular banding spacing increases with increased welding speed [28]. The semicircular banding spacing starts to appear at 300 mm/min and becomes noticeably wider at 500 mm/min welding speed, as shown in Figure 3.

The macrographs of the transverse cross-sections of the FSWed 5754-H24 aluminum alloy joints processed at a constant rotation rate of 600 rpm and different welding speeds (100–500 mm/min) using different tools are shown in Figure 4. Defect-free welded joints have been observed at all the applied welding conditions, except those processed using the tool with a pin eccentricity of 0.8 (the highest applied value) at both welding speeds of 300 and 500 mm/min. These two joints show tunnel defects. The tunnel size increases with increasing welding speed from 300 to 500 mm/min and tool pin eccentricity of 0.8 mm. The appearance of these tunnels is due to improper stirring in the NG zone because of low heat input. Increasing both welding speed [29] and pin eccentricities [30] decreases heat input in the NG zone. Many studies [17,31,32] have found that inadequate material movement around the pin during the stirring process as well as insufficient heat input in the NZ result in numerous defects such as kissing bond, cavities, and tunnel defects.

### 3.2. The Grain Structure and Texture of the As-Received AA5457-H24 Alloy

The grain structure of the as-received AA5457-H24 alloy has been investigated using the SEM-EBSD technique. Figure 5a,b presents the inverse pole figure (IPF), the coloring orientation image map (OIM), and the grain boundary (GB) map, with high-angle grain boundaries (HAGBs) >15° in black lines and low-angle grain boundaries (LAGBs) from 5° to <15° in red lines, with their corresponding grain size and misorientation angle distribution for the AA5754 as-received material. The grains are nearly equiaxed coarse grain structures that are dominated by mixed <001> red, <101> green, and <111> blue orientations. The GB map (Figure 5b) clearly shows that the boundaries are mainly of high angles with low density of low-angle boundaries, indicating that the structure is mainly dynamically recrystallized. The grain size varied, with very small fractions below 10 µm and large fractions above 10 up to 140 µm with an average grain size of 36 µm as observed from the grain size distribution (Figure 5c). The misorientation angle distribution (Figure 5d) shows the dominance of the HAGBs and the low fraction of LAGBs. Figure 5e depicts the 001, 101, and 111 pole figures for the EBSD data obtained for the as-received AA5457 alloy. The texture is a typical recrystallization texture of only three times random.

### 3.3. Grain Structure and Texture of the FSWed AA5754 Joints

#### 3.3.1. Effect of Tool Pin Eccentricity on Grain Structure

The grain structure and texture of the FSWed joints produced using the different tool pin eccentricities at a constant rotation rate of 600 rpm and two different welding speeds of 100 and 500 mm/min have been examined through the EBSD at the NG zone almost at the same position and using a step size of 1 µm. Figure 6 shows the IPF maps relative to the normal direction (ND) with their corresponding GB maps with the HAGBs > 15° in black lines and the 3° < LAGBs < 15° for the joints produced at 100 mm/min welding speed and different tool pin eccentricities of 0 (Figure 6a), 0.2 (Figure 6b), and 0.8 (Figure 6c). As depicted in Figure 6a, the grain structure of the map obtained for a 0-eccentricity joint mainly consists of fully recrystallized and equiaxed grains with a low density of LAGBs, which can be seen from the GB map. This is attributed to the dynamic recrystallization process during the FSW process due to the high strain and high temperature experienced [16,33]. The grain size varied from 3 to 60 µm with an average of 18 µm (Figure 7a). Clearly, a significant grain size reduction occurred in the NG zone after FSW. Using the pin eccentricity of 0.2 mm led to a larger reduction in the grain size distribution (Figure 7b) that varied from 3.2 µm to 46 µm and reduced the average grain size of 15 µm. Furthermore, an increase in the density of the substructure can be observed from the GB map (Figure 6b) and the misorientation angle distribution Figure 7b. This reduction of the grain size by applying 0.2 mm pin eccentricity can mainly be attributed to the lower temperature generated due to the longer rotational path of the material flow as reported by Essa et al. [34]. A similar observation is reported by Chen et al. [22] during the use of 0.4 mm pin eccentricity to FSWed AA5052 alloy. Also, from the IPF map in Figure 6b, the grains are slightly elongated (not fully equiaxed) and oriented at an angle of 45° relative to the ND. This can affect the longer material flow path around the pin. On the other hand, increasing the pin eccentricity to 0.8 mm has resulted in a fully equiaxed grain of slightly coarser than that obtained using 0.2 mm eccentricity. The grain size distribution (Figure 7c) ranges from 3.6 µm to 65 µm with an average of about 18 µm with a slightly lower density of substructures (Figure 6c) relative to that obtained using 0.2 mm pin eccentricity (Figure 6b). This can be explained by the increased heat input and high temperatures generated due to the excessively longer material flow path that affects the state of the material. Chen et al. [22] also reported this coarsening when increasing the pin eccentricity from 0.4 mm to 0.8 mm and attributed that to the change of material state to a fluid-like one that reduces the flow and material cooling behind the tool, resulting in a slightly coarser average grain size.

#### 3.3.2. Impact of Tool Welding Speed on the Grain Structure

The welding speed (traverse speed) significantly reflects the obtained grain size after FSW, as it affects the amount of heat generated during the process. Increasing the welding speed at a constant tool rotation rate decreases the heat generated and consequently reduces the final grain size [33,35]. This effect is observed in the current study, as two very different welding speeds of 100 and 500 mm/min were used at a constant tool rotation rate of 600 rpm. Figure 8 shows the IPF maps relative to the normal direction (ND) with their corresponding GB maps with the HAGBs > 15° in black lines and the LAGBs between 3° and 15° in red lines for the joints produced at 500 mm/min welding speed and different tool pin eccentricities of 0 mm (Figure 8a), 0.2 mm (Figure 8b), and 0.8 (Figure 8c) mm. It can be observed that increasing the welding speed from 100 to 500 resulted in a significant reduction in the average grain size from 18 µm to 12.4 µm for the 0 mm eccentricity, from 15 µm to 10 µm for the 0.2 mm eccentricity, and from 18 µm to 11 µm for the 0.8 mm eccentricity. This reduction in the average grain size by increasing the welding speed is mainly due to the relatively low heat generated and the relatively low amount of strain experienced after increasing the welding speed. Increasing the welding speed (from 100 to 500 mm/min) at a constant rotation rate of 600 rpm reduces the number of revolutions per mm in advance from 6 to 1.2 rev/mm. A similar behavior is reported by Ahmed et al. [33] when increasing the welding speed from 50 to 200 mm/min upon FSW of AA7075 alloy at a constant tool rotation rate of 300 rpm, which resulted in an average grain size reduction from 6 to 2 µm. They also described a similar behavior upon FSW of AA5083 alloy using the same welding conditions and noticed that the average grain size reduced from 9 to 3 µm [33]. Kumar et al. [36] studied the effect of the welding speed on the microstructure of AA2050 during FSW at a constant rotation rate of 600 rpm using different welding speeds from 60 mm/min up to 240 mm/min and reported a somewhat similar behavior of grain size reduction from 18 µm to 10.5 µm. They attributed this behavior to the lower heat input, which affects the deformation of the plasticized material [36]. Figure 9a–c shows the grain size distribution and the misorientation angle distribution calculated from the EBSD data presented in Figure 8. It can be observed that the grain size distribution is almost typically randomly distributed at all the different tool pin eccentricities, and the misorientation angle distributions have a somewhat similar distribution to that obtained at 100 mm/min welding speed. In comparison with the 100 mm/min welding speed, the grain size distribution range is lower here. For instance, in the joint produced using e = 0, the grain size distribution ranges from 3.2 to 45 µm with an average of about 12.4 µm. On the other hand, in the joint produced using e = 0.2, the grain size distribution ranges from 3.2 to 45 µm with an average of 10 µm. The joint produced using e = 0.8, the grain size distribution ranges from 2.9 to 35 µm with an average of 11 µm.

#### 3.3.3. Crystallographic Texture Evolution

The crystallographic texture was investigated at the NG zone of the AA5457-H24 alloy welded using different pin eccentricities. Figure 10 shows the 001, 101, and 111 pole figures calculated from the EBSD data presented in Figure 6 for the NG zone of the joints produced at 100 mm/min and tool pin eccentricities of 0 (a), 0.2 (b), and 0.8 mm (c). It should be mentioned here that the EBSD data were rotated to align the FSW reference frame (WD, TD, and ND) with the shear reference frame (Ө, z, and r), which causes the alignment of intense (111) pole with the shear plane normal [37]. FSW is a severe plastic deformation process in which the main deformation mechanism is shear deformation. Shear deformation is one of the primary mechanisms that occur during FSW. As the rotating tool penetrates the workpiece, it generates heat due to friction between the tool and the workpiece. This heat softens the material, causing it to deform plastically. The tool then stirs the softened material, causing it to undergo shear deformation as it is pushed aside and moved around the tool. The shear reference frame is not fixed during FSW. Still, it rotates as the tool rotates, resulting in a misalignment between the shear reference frame and the FSW reference frame that reflects directly on the simple shear texture components being off axes in the raw EBSD data. It needs specific rotations to obtain the texture components at their ideal positions. Table 3 shows the different types of texture fibers that can develop upon shear deformation depending on the extent of strain experienced. After applying the required rotations, it can be observed that the texture is dominated by $\bar{B}$ {$\bar{112}$} <1$\bar{1}$0> simple shear texture component with some C {$\bar{001}$} <1$\bar{1}$0> simple shear texture component that can be observed in the pole figures depicted in Figure 10 for e = 0 (a) and e = 0.2 mm (b). These simple shear texture components of $\bar{B}$ and C can also be observed in the ODF contour sections at constant φ2 = 0° and φ2 = 45°, with the ideal texture components are superimposed at their ideal positions in both ODF sections, for the same data obtained from the NG of the joints produced at 100 mm/min and different pin eccentricities of e = 0 (Figure 11a), e = 0.2 (Figure 11b), and e = 0.8 mm (Figure 11c). It was also detected that the texture intensity is about 6 times random at e = 0, which reduced to about 5 at e = 0.2 and slightly increased again to about 5.3 times random at e = 0.8. The pole figures in Figure 10c (for e = 0.8) describe the exitance of the B texture component {11$\bar{2}$} <1$\bar{1}$0>, and the C component was also observed in both the pole figures and the ODF section in Figure 11. From the ODF sections at φ2 = 0° (C components) and φ2 = 45° (B/$\bar{B}$ and A components) in Figure 11a, it can be observed that the texture components are almost at their ideal positions as they nearly match the ideal positions indicated; however, the ODF sections of the samples with e = 0.2 and e = 0.8 (Figure 11b,c) texture components are located away from their ideal positions, as can be noted. This implies that the eccentricity added more complexity to the shear deformation path, and the applied rotations to compensate for the tool’s rotation and tapering are not enough to align the shear deformation reference frame with the deformation reference frame.

Figure 12 shows the 001, 101, and 111 pole figures calculated from the EBSD data presented in Figure 8 for the NG zone of the joints produced at 500 mm/min and various pin eccentricities of 0 (Figure 12a), 0.2 (Figure 12b), and 0.8 mm (Figure 12c). Again, the pole figures are presented after applying the required rotations to align the deformation reference frame with the shear reference frame. The ODF contour sections at constant φ2 = 0° and φ2 = 45° are depicted in Figure 13 for pin eccentricities of 0 (Figure 13a), 0.2 (Figure 13b), and 0.8 mm (Figure 13c) with the ideal texture components superimposed at their ideal positions in both ODF sections. It can be observed that the texture is a simple shear texture with both B/$\bar{B}$ texture component with C component at their ideal positions in both the pole figures and ODF sections in case of the 0 pin eccentricity (Figure 13a), whereas the ODF sections of the samples with pin eccentricities of 0.2 (Figure 13b) and 0.8 (Figure 13c) again show clear texture components’ deviation from the ideal positions, as can be observed. This can also be attributed to the complex deformation path with the existence of tool pin eccentricity.

### 3.4. Tensile Strength and Weld Joint Efficiency

Stress-strain curves of the as-received AA5754-H24 alloy and the FSWed joints at a constant rotation of 600 rpm, various welding speeds of 100, 200, 300, and 500 mm/min, and several eccentricities of 0.0, 0.2, and 0.8 mm were determined and presented in Figure 14. The engineering tensile stress-strain curves of the FSWed joints using a normal tool without pin eccentricity (e = 0) are shown in Figure 14a. The as-received AA5754-H24 alloy is strain hardened and then partially annealed (half-hard) so that it shows higher yield stress, ultimate tensile stress, and a prolonged curve until fracture. This gives the as-received AA5754-H24 alloy higher toughness, which can be indicated by the large area under the stress-strain curve, than the FSWed samples. The stress-strain curves show repeated serration, known as the Portevin-Le Chatelier (PLC) effect [40,41,42], which is related to the subsequent pinning and unpinning of the moving dislocations by the solute atoms. It was noticed that the whole FSWed samples have lower tensile properties (yield stress, ultimate tensile stress, and fracture strain) than the as-received alloy. The FSWed sample welded at a higher travel speed of 500 mm/min showed the highest strength of all the welded samples, while the other samples welded at 100, 200, and 300 mm/min show nearly comparable tensile behavior. The stress-strain curves of the FSWed joints with e = 0.2 mm are depicted in Figure 14b. Pin eccentricity and a high travel speed of 500 mm/min have produced a joint with higher strength than the as-received alloy within the strain range of the welded sample (up to ε = 5.5%). The increased strength of the welded samples at 500 mm/min has decreased joint ductility. Welded material at the lowest speed (100 mm/min) showed the lowest tensile strength and the highest elongation compared with the other welded samples under the same welding conditions, yet still lower than the as-received alloy. Figure 14c describes that all FSWed samples at a pin eccentricity of 0.8 mm have lower tensile properties than the as-received alloy. Nevertheless, none of the welded samples gave a clear variation in tensile deformation behavior with the different travel speeds, where the stress-strain curves are gathered in a bundle with nearly similar behavior.

The tensile strength of the FSWed joints produced using various tool pin eccentricities is plotted against the welding speeds, as depicted in Figure 15a. It was noticed that the tensile strength and the yield stress of the FSWed joints welded with the tool pin eccentricities of 0.2 and 0.8 mm are generally higher than those welded without tool pin eccentricity (e = 0). Moreover, the tensile strength of the joints welded with e = 0.2 mm is higher than that welded using e = 0.8 mm, as shown in Figure 15a. In addition, there is a slight increase in the tensile strength with increasing welding speed, which is more likely related to the lower heat input at higher speeds [43]. Furthermore, the defect-free FSWed joints produced using a tool with e = 0.2 mm shows the highest tensile strength at a higher welding speed of 500 mm/min, as described in Figure 14b. Compared with the tool without pin eccentricity, the tool with e = 0.8 mm has shown improved strength at lower welding speeds (100 and 200 mm/min), which changes to a slight decrease at the higher speeds of 300 and 500 mm/min due to the formed welding defects in the form of tunnels at the root of the weld NG, as depicted in Figure 4. This decrease in the tensile strength was also related to the stress concentration formed by a higher heat input rate, which forms a worse interface where crack initiation becomes possible, resulting in a significant decrease in the mechanical properties of the weld joint, as noted, at an eccentricity over 0.3 mm [18,44]. At low travel speeds (100 and 200 mm/min), the yield stress (σ_0.2%_) of the FSWed joints produced using the high eccentricity tool (e = 0.8 mm) was markedly higher than that of the other joints as presented in Figure 15a.

Furthermore, at the welding speed of 500 mm/min, the yield stress increase is much more noticeable than the improvement in the tensile strength at the same speed. The strain at fracture of the whole FSW joints was much lower than that of the as-received alloy, as depicted in Figure 15b. Generally, the tensile strength of the FSWed joints that is lower than the as-received alloy accelerates the change from the homogeneous plastic deformation over the test samples to localized deformation at the weld NG, which shortens the strain till fracture. Nevertheless, FSW with tool pin eccentricity further decreases the degree of deformation before fracture.

The weld joint efficiency expressed as the tensile strength of the FSWed joint relative to that of the as-received alloy is presented in Figure 16a. The highest weld joint efficiency was obtained at e = 0.2 mm at all welding speeds, except for the weld joint produced at the welding speed of 100 mm/min, which showed further increase at e = 0.8 mm. On the one hand, the weld joint efficiency of 97% was attained at 500 mm/min for the pin eccentricity of 0.2 mm. On the other hand, at the pin eccentricity of 0.8 mm, the weld joint efficiency is significantly decreased at 300 and 500 mm/min welding speeds. This is attributed to the realized defects (as shown in Figure 4) at these welding conditions. These results agree with those acquired by Elangovan et al. [18] and Yu Chen et al. [22] for FSW of AA2219 and AA5052 alloys, respectively. Figure 16b describes the hardness distribution over the cross-section of the FSWed joints at the welding speed of 500 mm/min and several tool pin eccentricities. The typical FSW hardness profiles for the NG zone and the various affected regions have been detected. In agreement with the results of tensile properties, the highest hardness measurements of the NG were obtained for the FSWed joints welded using the tool with e = 0.2 mm. In contrast, the lowest hardness values were shown to be changeable between the welded joints using e = 0 and e = 0.8 mm. The hardness behavior of the W shape profile indicates a reduction in the hardness in the weld zone with a slight increase in the SZ. This can mainly be attributed to the starting condition of the alloy being strain hardened, and the experience of the high thermal cycle during FSW results in softening by reducing the number of dislocations and the different types of imperfections, such as point defects and the substructure. It has been reported that the hardness traverses in work-hardened non-heat-treatable alloys (e.g., 5xxx alloys in the H1xx, H2xx, or H3xx conditions) normally resemble the W shape profile similar to that shown in Figure 16b. As the weld is accomplished, the heat from the FSW process causes annealing and recovery to occur, leading to a drop in the hardness [45].

## 4. Conclusions

The impact of the FSW tool pin eccentricity (e = 0, 0.2, and 0.8 mm) and a wide range of welding speeds (100, 200, 300, and 500 mm/min) on the grain structure, texture, and mechanical behavior of the FSWed AA5754-H24 alloy were investigated, and based on the acquired results, the following conclusions can be drawn:The NG macro investigations showed defect-free joints at pin eccentricities of 0 and 0.2 mm using all welding speeds. In contrast, at the pin eccentricity of 0.8 mm, tunnel defects appear at welding speeds of 300 and 500 mm/min.The grain size in the NZ of all the processed specimens at the suggested welding conditions in terms of welding speeds (100–500 mm/min), different pin eccentricities (0, 0.2, and 0.8 mm), and applying a constant tool rotation rate of 600 rpm is lower than that given by the as-received AA5754-H24 alloy.Grain structure studied using EBSD for the NG of the joints produced at 100 mm/min, 600 rpm, and different tool pin eccentricities showed significant grain refining due to dynamic recrystallization with an average grain size of 18 µm at e = 0, and 0.8 mm, and 15 µm at e = 0.2 mm. Increasing the welding speed from 100 to 500 mm/min further reduced the average grain size of the NG zone to 12.4, 10, and 11 µm at e = 0, 0.2, and 0.8 mm, respectively.The tool pin eccentricity of (e = 0.2 mm) caused the smallest average grain size at 100 and 500 mm/min of 15 and 10 µm, respectively.The texture is found to be dominated by the simple shear texture, with both the $\bar{B}$/B texture component and C component at their ideal positions after rotating the data to align the shear reference frame with the FSW reference frame in both pole figures and ODF sections.The tensile properties for the FSWed joints were slightly lower than the as-received alloy due to the hardness reduction in the weld zone. On the other hand, the tensile strength and yield stress for all joints increased by increasing the FSW speed from 100 to 500 mm/min.Welding using the pin eccentricity of 0.2 mm resulted in the highest tensile strength; at a welding speed of 500 mm/min, it reached 97% of the strength of the as-received alloy.The fracture strains for the whole FSWed samples were much lower than that of the base alloy. Moreover, the FSWed joints produced using a tool with pin eccentricity showed a further decrease in fracture strain, especially at e = 0.2 mm, due to the increased strength.The hardness profile showed the typical W shape with a reduction in the hardness of the weld zone and a slight recovery of the hardness in the NG zone.

## Figures and Tables

**Figure 1 materials-16-02031-f001:**
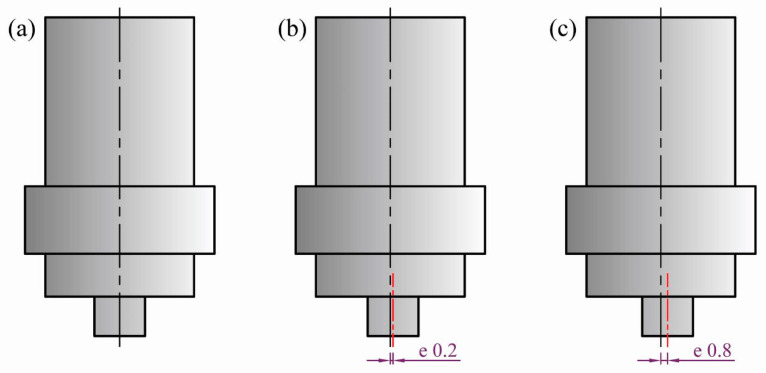
Schematic drawing of tool pin eccentricity, (**a**) e = 0, (**b**) e = 0.2, and (**c**) e = 0.8 mm.

**Figure 2 materials-16-02031-f002:**
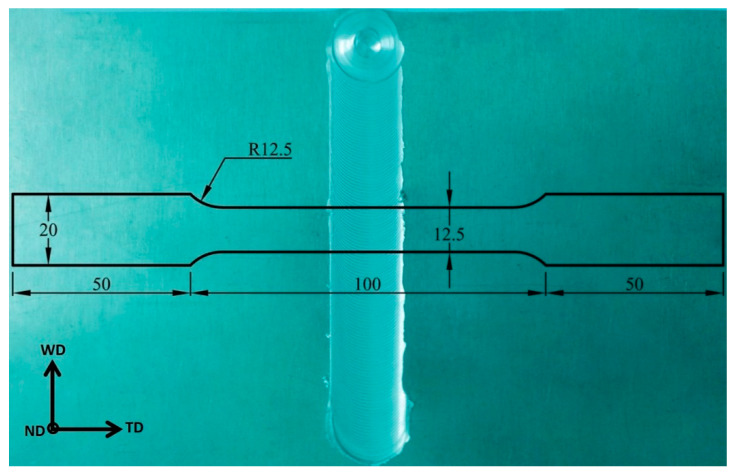
Tensile test specimen with all dimensions in mm superimposed on the top surface of the FSWed sample at a rotation rate of 600 rpm and a welding speed of 200 mm/min using a pin eccentricity of 0.2. Note: WD: welding direction, TD: transverse direction, and ND: normal direction.

**Figure 3 materials-16-02031-f003:**
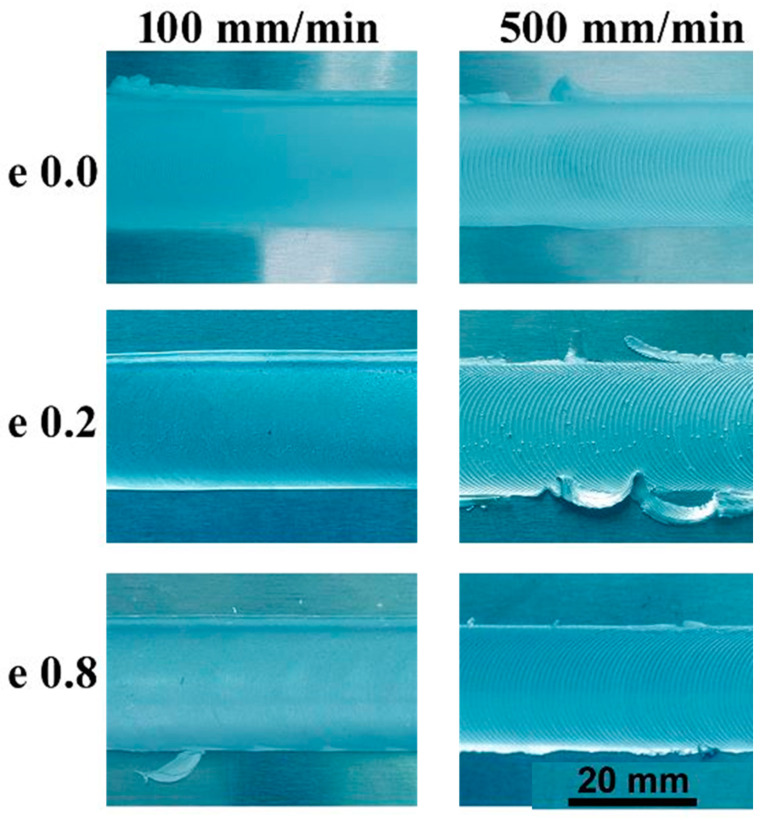
Top surface view of AA5754-H24 FSWed joints at a rotation speed of 600 rpm, welding speeds of 100 and 500 mm/min, and applying different pin eccentricities of 0, 0.2, and 0.8 mm.

**Figure 4 materials-16-02031-f004:**
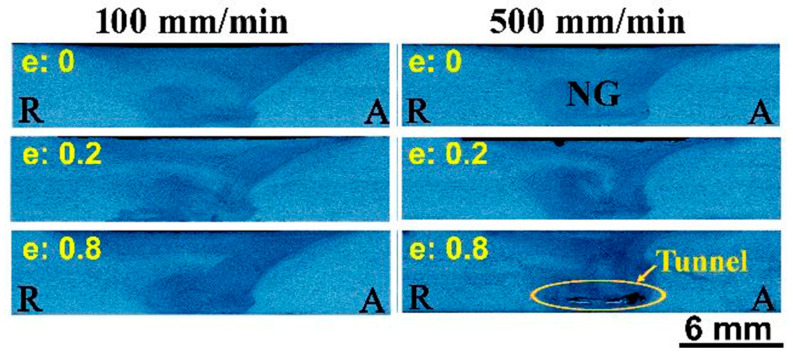
Transverse cross-section macrographs of AA5754-H24 FSWed Joints welded at welding speeds (100 and 500 mm/min) and different eccentricities (e = 0, e = 0.2, and e = 0.8 mm) using a constant rotation rate of 600 rpm.

**Figure 5 materials-16-02031-f005:**
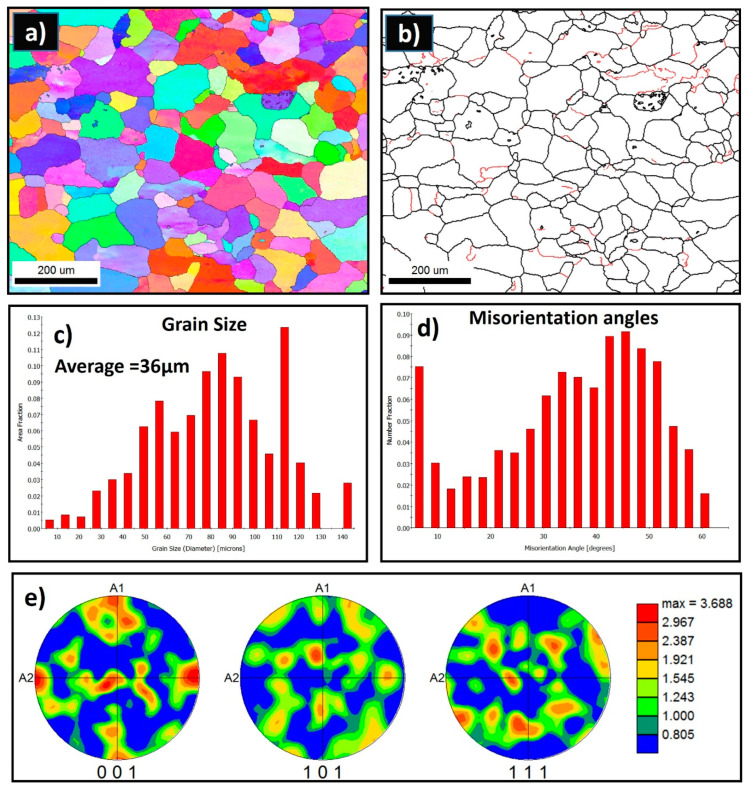
(**a**) IPF map and (**b**) GB map with HAGBs > 15° in black lines and 15° > LAGB’s > 5° in red lines, with their corresponding grain size (**c**), misorientation angle (**d**) distribution, and pole figures (**e**) for the AA5754 alloy acquired at 2 µm step size.

**Figure 6 materials-16-02031-f006:**
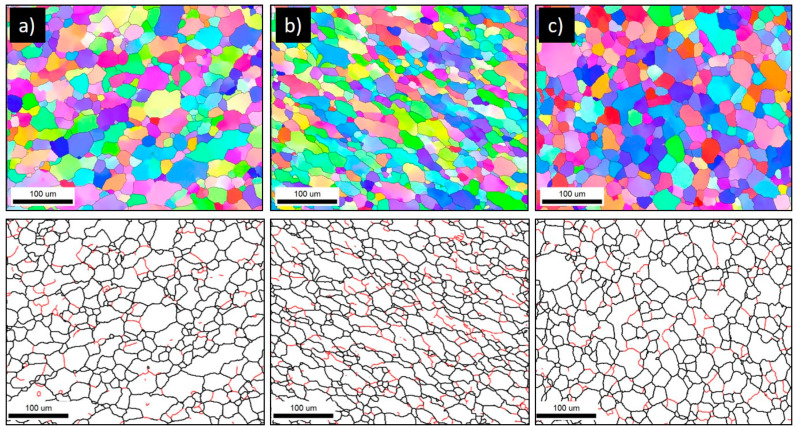
IPF and GB maps of the FSWed AA5754-H24 at a constant rotation rate of 600 rpm and constant welding speed of 100 with different tool pin eccentricities: (**a**) e = 0, (**b**) e = 0.2, and (**c**) e = 0.8 mm. The EBSD data were acquired at 1 µm step size.

**Figure 7 materials-16-02031-f007:**
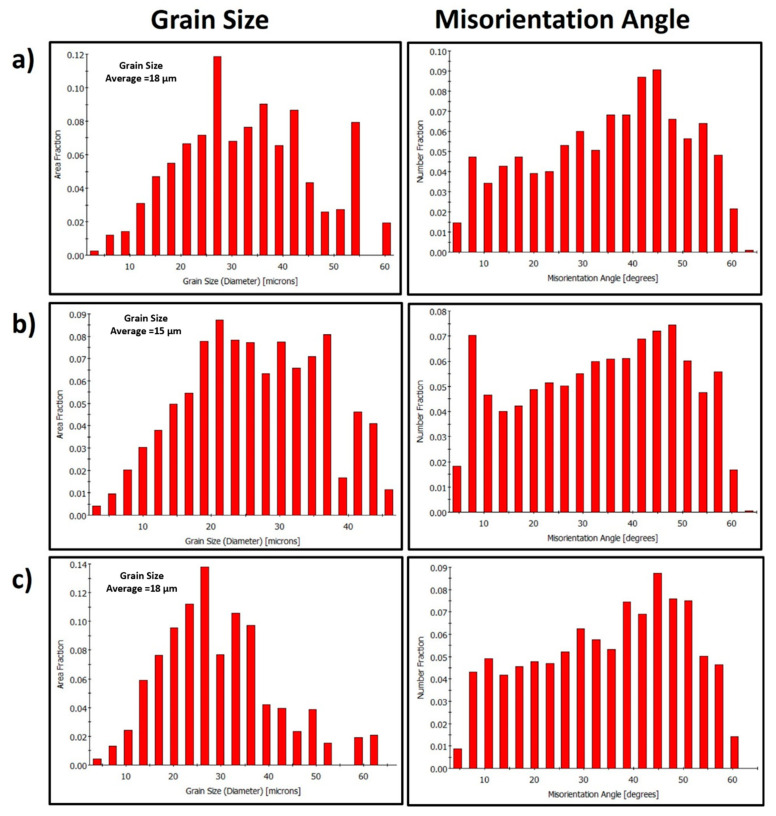
Grain size and misorientation angle distributions calculated from the EBSD data presented in Figure 6 and obtained from the NG zone of the FSWed AA5457-H24 alloy using a constant welding speed of 100 mm/min, a tool rotation speed of 600 rpm, and different tool pin eccentricities: (**a**) e = 0, (**b**) e = 0.2, and (**c**) e = 0.8 mm.

**Figure 8 materials-16-02031-f008:**
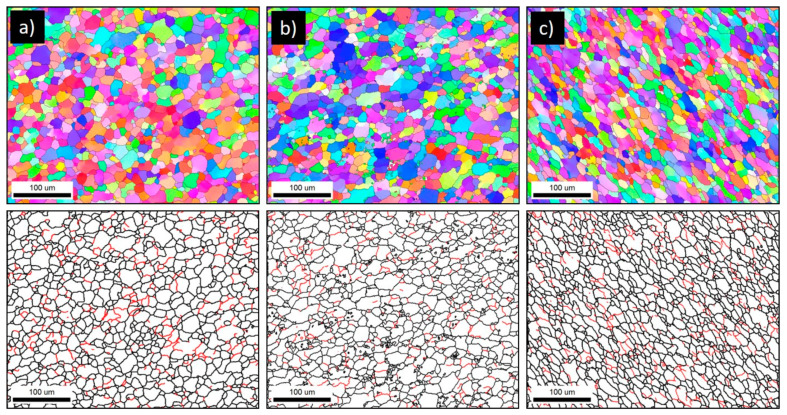
IPF and GB maps of the FSWed AA5754-H24 alloy at a constant rotation rate of 600 rpm and a welding speed of 500 mm/min, applying several tool pin eccentricities: (**a**) e = 0, (**b**) e = 0.2, and (**c**) e = 0.8 mm. The EBSD data were acquired at 1 µm step size.

**Figure 9 materials-16-02031-f009:**
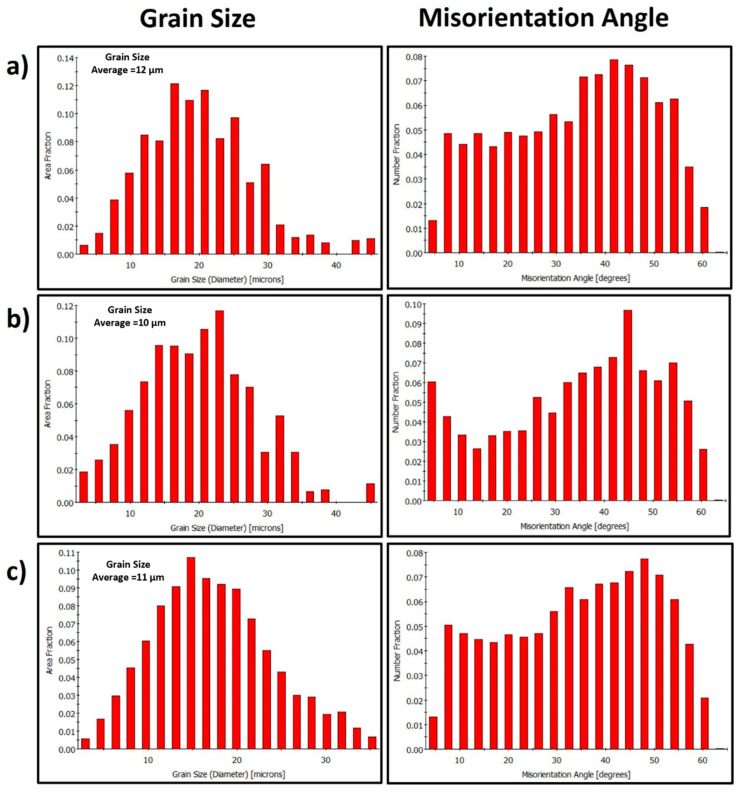
Grain size and misorientation angle distributions calculated from the EBSD data presented in Figure 8 and obtained from the NG zone of the FSWed AA5457-H24 alloy using a constant welding speed of 500 mm/min and a rotation speed of 600 rpm at various tool pin eccentricities: (**a**) e = 0, (**b**) e = 0.2, and (**c**) e = 0.8 mm.

**Figure 10 materials-16-02031-f010:**
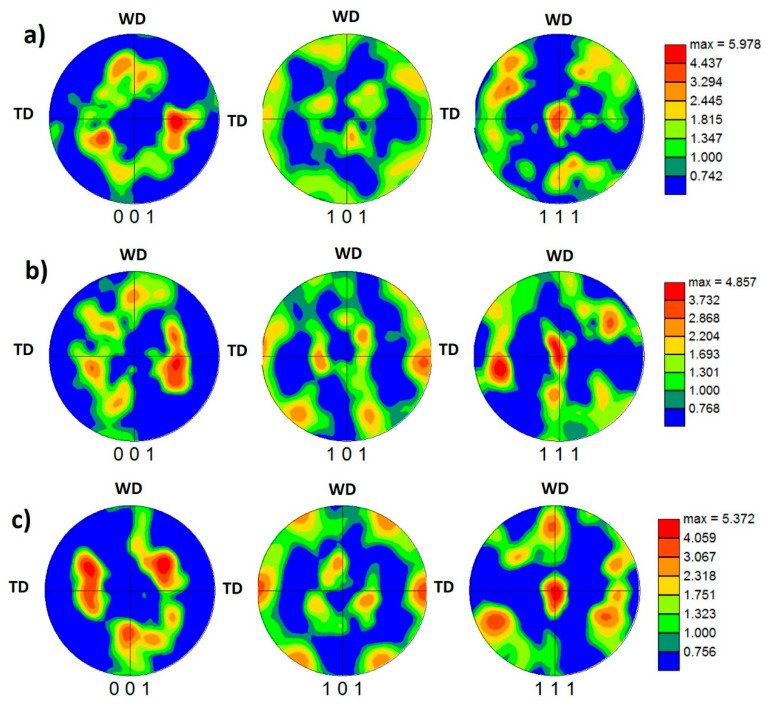
The 001, 101, and 111 pole figures calculated from the EBSD data presented in Figure 6 for the NG zone of the joints produced at 100 mm/min and pin eccentricities of (**a**) e = 0, (**b**) e = 0.2, and (**c**) = 0.8 mm.

**Figure 11 materials-16-02031-f011:**
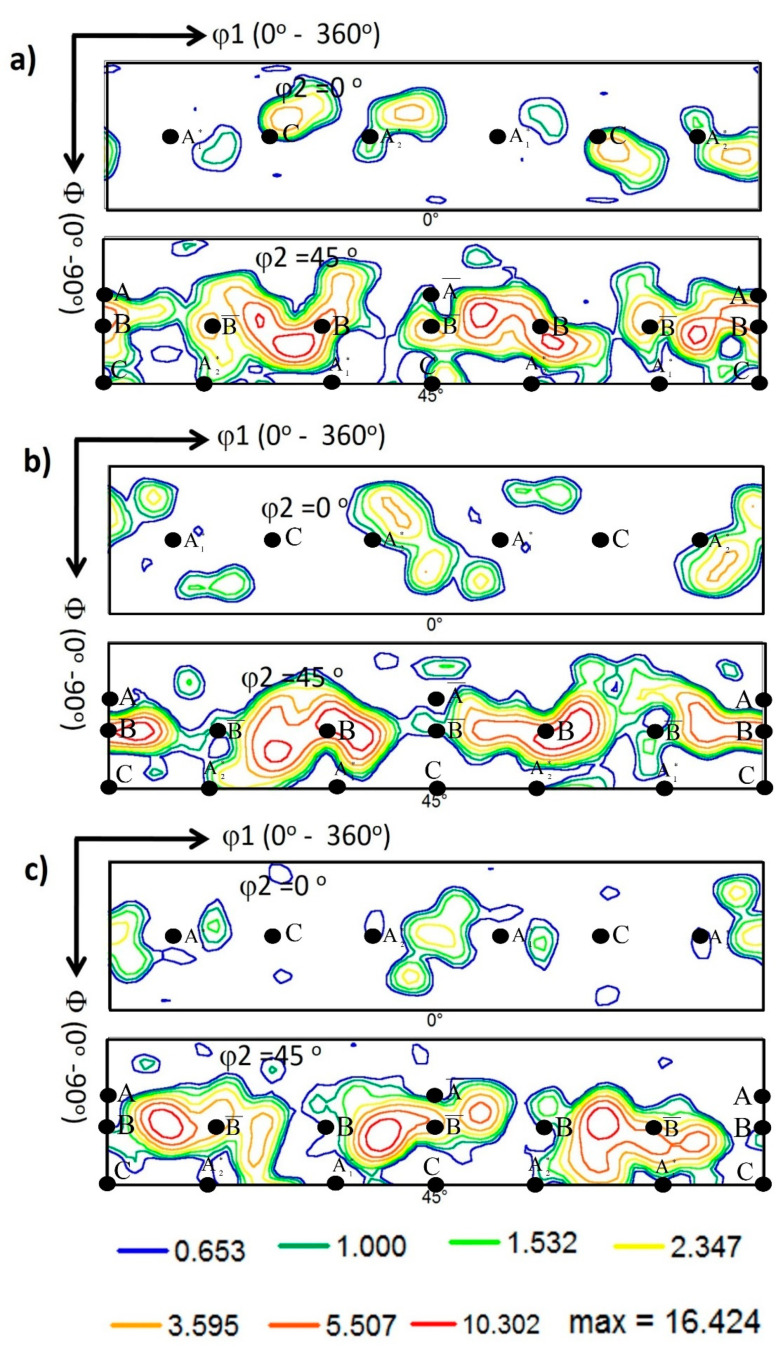
ODF contour sections at constant φ2 of 0° and 45° calculated from the EBSD data presented in Figure 6 for the NG zone of the joints produced at 100 mm/min and different pin eccentricities of (**a**) e = 0, (**b**) e = 0.2, and (**c**) = 0.8 mm. The coloring key is presented below the Figure. The ideal texture components are superimposed at their ideal positions in both ODF sections.

**Figure 12 materials-16-02031-f012:**
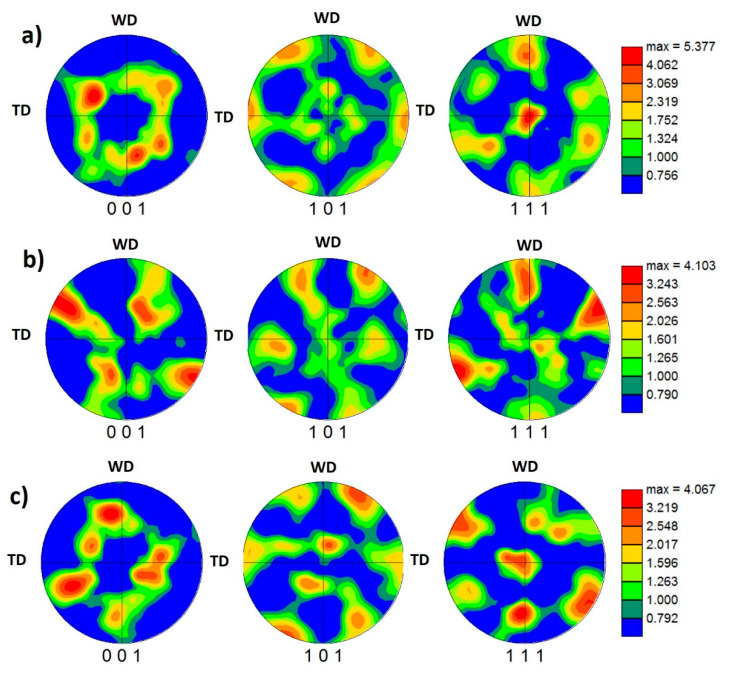
The 001, 101, and 111 pole figures calculated from the EBSD data presented in Figure 8 for the NG zone of the joints produced at 500 mm/min and pin eccentricities of (**a**) e = 0, (**b**) e = 0.2, and (**c**) = 0.8 mm.

**Figure 13 materials-16-02031-f013:**
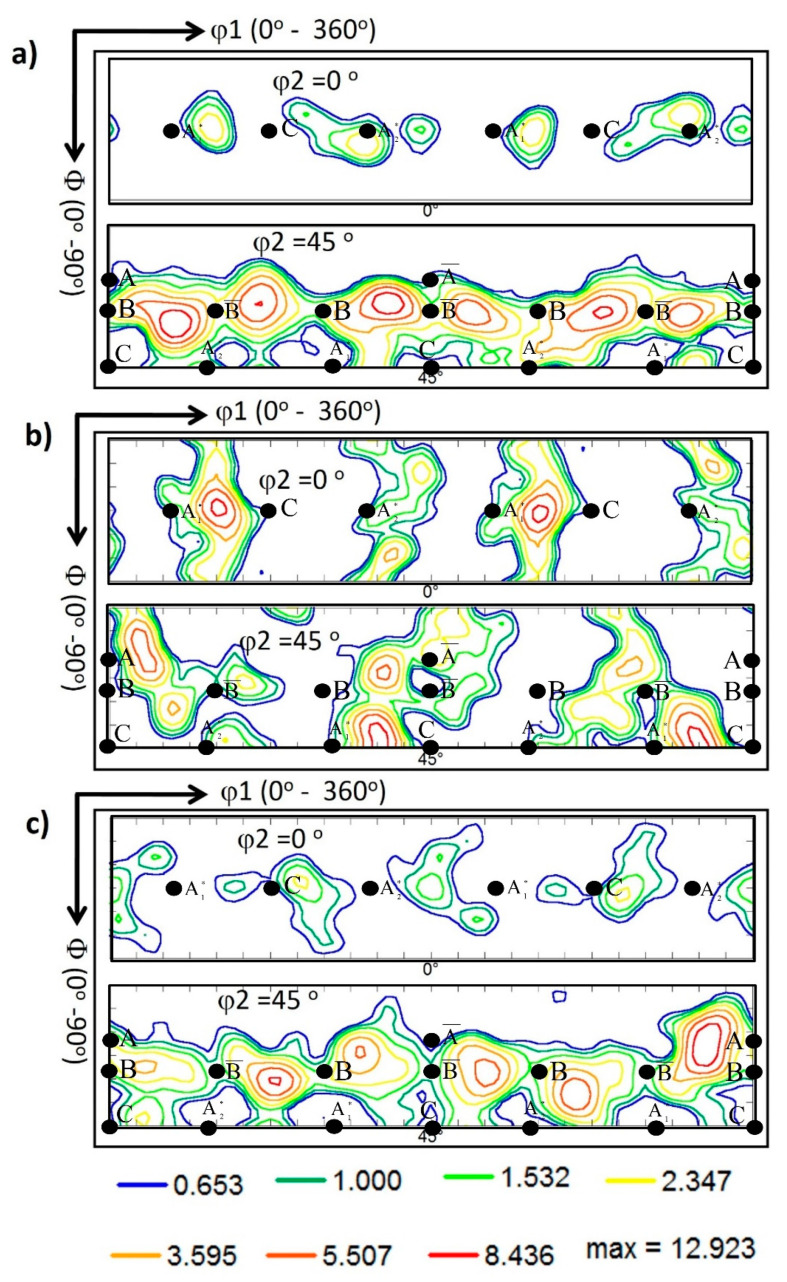
ODF contour sections at constant φ2 of 0° and 45° calculated from the EBSD data presented in Figure 8 for the NG zone of the joints produced at 500 mm/min and different pin eccentricities of (**a**) e = 0, (**b**) e = 0.2, and (**c**) = 0.8 mm. The coloring key is presented below the Figure. The ideal texture components are superimposed at their ideal positions in both ODF sections.

**Figure 14 materials-16-02031-f014:**
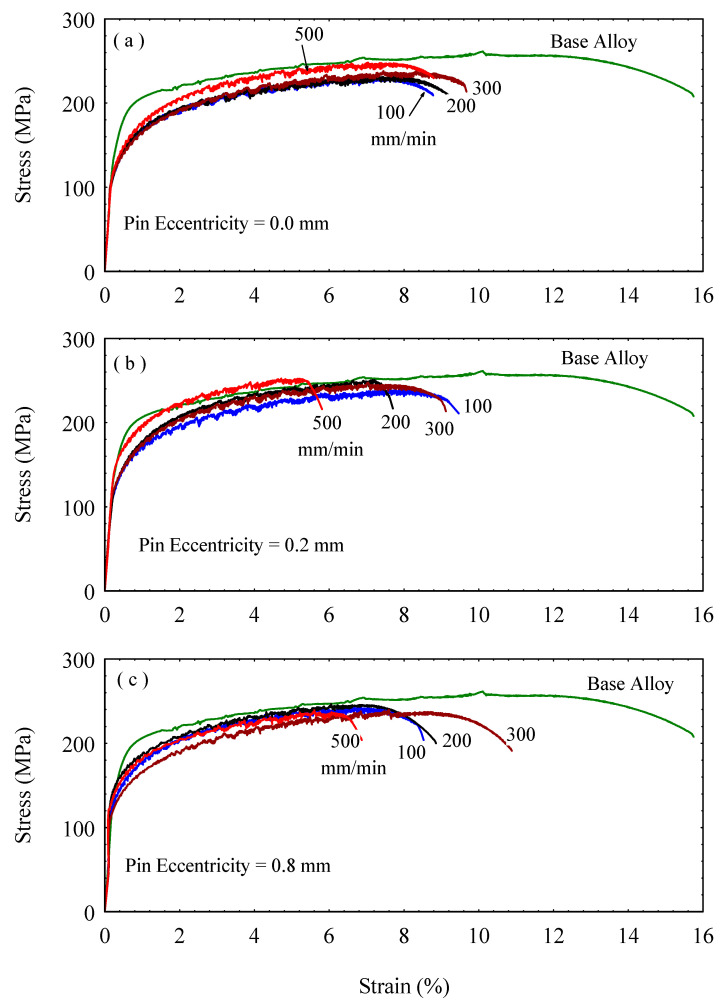
Stress-strain curves of 5754-H24 FSW joints at tool pin eccentricities of (**a**) e = 0, (**b**) e = 0.2, and (**c**) = 0.8 mm.

**Figure 15 materials-16-02031-f015:**
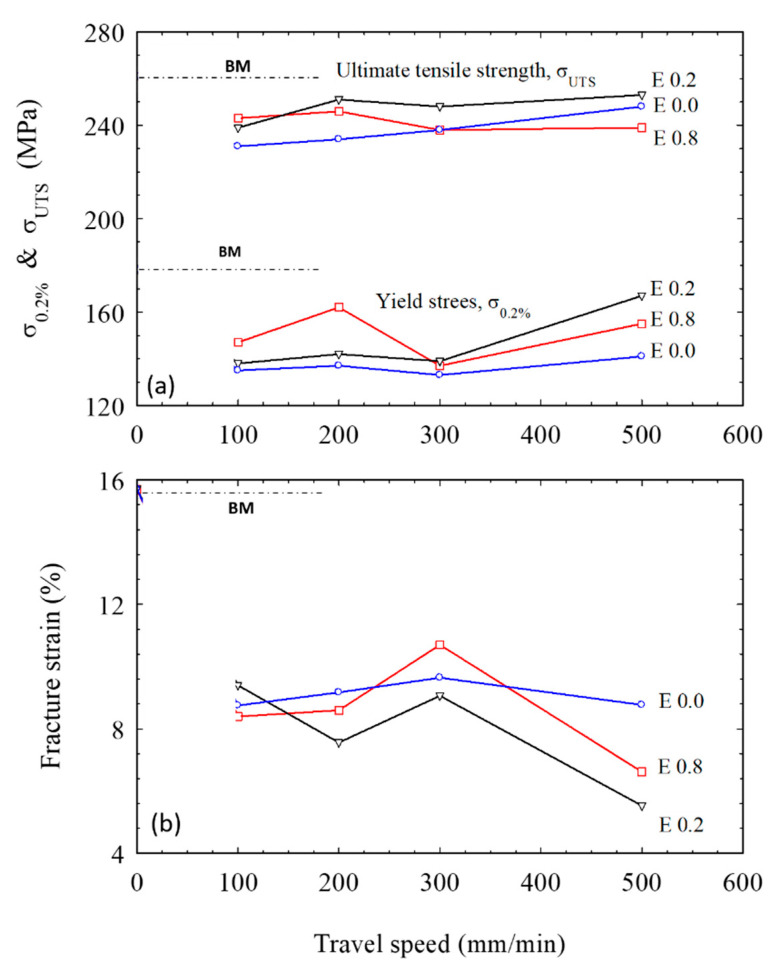
Comparison of the tensile properties of AA5754-H24 FSWed joints at various pin eccentricities (e = 0, 0.2, and 0.8 mm).

**Figure 16 materials-16-02031-f016:**
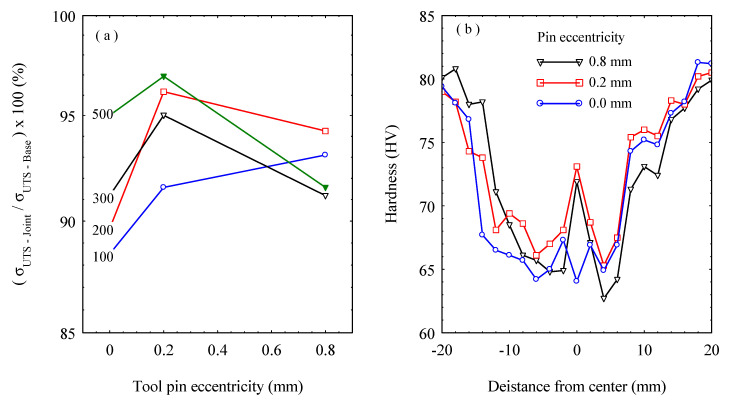
(**a**) Weld joint efficiency against the tool pin eccentricities at various welding speeds; (**b**) Vickers hardness profiles over cross-section of AA5754-H24 FSWed joints at a welding speed of 500 mm/min.

**Table 1 materials-16-02031-t001:** The chemical composition of AA5754-H24 alloy.

wt.%								
Si	Fe	Cu	Mn	Mg	Cr	Zn	Ti	Al
0.4	0.4	0.1	0.5	2.6-	0.3	0.2	0.15	Rest

**Table 2 materials-16-02031-t002:** The mechanical properties of AA5754-H24 alloy.

Tensile Strength (MPa)	Proof Stress 0.2% (MPa)	Elongation (%)
261 ± 3	178 ± 3	15.7 ± 2

**Table 3 materials-16-02031-t003:** Partial fibers and ideal components of simple shear texture in fcc metals [38,39].

A/B Fiber	Shear Plane	Shear Direction	Euler Angles (°)		
	(hkl)	<uvw>	φ_1_	Φ	φ_2_
$A_{1}^{*}$	$\left\{11\bar{1}\right\}$	$\langle 2\bar{1}1\rangle$	35.26/215.26	45	0/90
			125.26	90	45
$A_{2}^{*}$	$\left\{11\bar{1}\right\}$	〈112〉	144.74	45	0/90
			54.74/234.74	90	45
$\bar{A}$	$\left\{11\bar{1}\right\}$	$\langle 1\bar{1}0\rangle$	0	35.26	45
$\bar{A}$	$\left\{11\bar{1}\right\}$	〈101〉	180	35.26	45
B	{112}	$\langle 1\bar{1}0\rangle$	0/120/240	54.74	45
$\bar{B}$	$\left\{\bar{1}\bar{1}\bar{2}\right\}$	$\langle 1\bar{1}0\rangle$	60/180	54.74	45
C	{001}	$\langle 1\bar{1}0\rangle$	90/270	45	0/90
			0/180	90	45

## Data Availability

Data will be available upon request through the corresponding author.

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
