# Peer review of "Friction Stir Welding of AA5754-H24: Impact of Tool Pin Eccentricity and Welding Speed on Grain Structure, Crystallographic Texture, and Mechanical Properties"

_materials, 2023, doi:10.3390/ma16052031_

Round 1
Reviewer 1 Report
The paper studied the effect of process parameters of friction stir welding on the grain structure, crystallographic texture, and mechanical properties of AA5754-H24. And some research results are obtained. The research results have certain value and significance. However, the paper has the following deficiencies, which need to be corrected and supplemented before the final acceptance.
1. How many specimens were tested for these properties, like tensile testing and Vickers hardness, and the test data should be marked with error bars in Figures 15 and 16.
2. The author has obtained rich research data, but the analysis of these data is not deep enough, and many places are simply listed. For example, the change curve of microhardness, why there is such a change, it is suggested to deepen the depth of theoretical analysis.
3. The language of the paper is not concise enough and needs to be refined.
4. Why design three tool pin eccentricities of 0, 0.2 and 0.8 mm? If the theoretical support behind can be provided in the text, it will improve the readability of the paper. From the results, the influence of this factor is not obvious.
Author Response
Response to comments of Reviewer 1
Dear Reviewer,
We do appreciate your positive feedback and the constructive comments you have provided that improved our manuscript significantly.
The paper studied the effect of process parameters of friction stir welding on the grain structure, crystallographic texture, and mechanical properties of AA5754-H24. And some research results are obtained. The research results have certain value and significance. However, the paper has the following deficiencies, which need to be corrected and supplemented before the final acceptance.
Comment #1
- How many specimens were tested for these properties, like tensile testing and Vickers hardness, and the test data should be marked with error bars in Figures 15 and 16.
Response
Many thanks for the valuable comment. Only one specimen was tested for each condition, either for the tensile properties or the hardness. Especially in the hardness test, a profile is obtained, so repeating the measurements will not help as we are not getting the average values. Regarding the tensile test, we agree it is important to carry the test for at least three samples from each condition; however, this was not available due to the limitation in the samples. Also, the tensile results look consistent and reflect the parameters
Comment #2
- The author has obtained rich research data, but the analysis of these data is not deep enough, and many places are simply listed. For example, the change curve of microhardness, why there is such a change, it is suggested to deepen the depth of theoretical analysis.
Response
Many thanks for the valuable comment. This important recommendation has been considered, and a more in-depth analysis of the results has been added to the revised manuscript. The hardness section has been improved, and more explanation is given.
Comment #3
- The language of the paper is not concise enough and needs to be refined.
Response
Many thanks for the critical comment. The language of the article has been revised thoroughly as per recommendation.
Comment #4
- Why design three tool pin eccentricities of 0, 0.2 and 0.8 mm? If the theoretical support behind can be provided in the text, it will improve the readability of the paper. From the results, the influence of this factor is not obvious.
Response
Many thanks for the comment. Based on the literature investigation, two parameters have been examined in this work, and the tool geometry is one of the important parameters affecting the heat generation and microstructure of the weld zone. Thus, we consider this an important parameter to examine with the traverse speed to enrich and add to the existing studies. Also, we see some critical effects for this parameter that we will highlight more in the revised manuscript.

Reviewer 2 Report
What is the reason behind choosing a constant tool rotational speed? On what basis the authors have chosen 600 rpm? Report with evidence.
Section 2.1,Coin a different name for homemade machine. Must be a modified milling machine.....
In EBSD why the authors intended to chose different step size for the as-received material and processed ones?
Pg 8/22, line 209 the word "pen" should be indicated as "pin". Must be a typographic error. Check for same errors in the following paragraphs.
On visual observation of Fig. 6 a&c, Fig.6c displays less coarse grains than the microstructure shown in Fig 6a. However, in the text it is represented in the reverse manner.
Section 3.3.1 must be proofread as some sentences are in concise
The authors presented the results obtained from texture analysis. However, the discussion is lagging. The role of plastic deformation on the preferred orientation should be discussed with their individual textural components
Say, Cube, Bs, Cu, Dillamore and so on...
Hardness is decreasing but UTS is increasing. Explain.
On addressing the above comments the paper can please be accepted for its publication
Author Response
Response to comments of Reviewer 2
Dear Reviewer,
We do appreciate your positive feedback and the constructive comments you have provided that improved our manuscript significantly.
Comment #1
What is the reason behind choosing a constant tool rotational speed? On what basis the authors have chosen 600 rpm? Report with evidence.
Response
Many thanks for the question. We have carried out many preliminary experiments for FSW of this alloy at our lab and found this rpm is suitable for obtaining sound joints at a wide range of traverse speeds. We have been interested in joining at very high speed, for example, at 500 mm/min, and using this rpm, we achieved this high welding speed for this alloy.
Comment #2
Section 2.1, Coin a different name for homemade machine. Must be a modified milling machine.....
Response
Thanks for the comment. Actually, it is really homemade (it means we have designed and manufactured a gantry-type FSW machine with local collaborators in design and manufacturing). If you check this video on youtube (https://www.youtube.com/watch?v=JZA7LQjF1hE ) you can see our FSW machine. Also it is pictured in the following reference.
(Elfishawy, E.; Ahmed, M.M.Z.; El-Sayed Seleman, M.M. Additive Manufacturing of Aluminum Using Friction Stir Deposition. In Proceedings of the TMS 2020 149th Annual Meeting {\&} Exhibition Supplemental Proceedings; Springer International Publishing: Cham, 2020; pp. 227–238.)
Comment #3
In EBSD why the authors intended to chose different step size for the as-received material and processed ones?
Response
Thanks for this critical question. The difference in the grain size between the BM and the processed material is clear, and the choice of the step size in EBSD depends on the expected grain size. Thus if we use a small step size in the BM we might end up with very small number of grains that can not represent the full features of the microstructure and texture. Also, EBSD data acquisition is quite time-consuming, so it is important to compromise between a number of factors in choosing the step size for EBSD analysis.
Comment#4
Pg 8/22, line 209 the word "pen" should be indicated as "pin". Must be a typographic error. Check for same errors in the following paragraphs.
Response
Many thanks for the notice. We have corrected this mistake throughout the article, and yes, it is a typo error, but, unfortunately, repeated. Many thanks again.
Comment#5
On visual observation of Fig. 6 a&c, Fig.6c displays less coarse grains than the microstructure shown in Fig 6a. However, in the text it is represented in the reverse manner.
Response
Thanks for the observation. As per the findings, the tool without eccentricity (e=0.0) results in an average of 18 µm grain size, and this is reduced to about 12µm using e= 0.2 where the use of pin eccentricity of e=0.8 has resulted in an average grain size of about 18 µm again with a slight increase. Initially, these values are calculated based on the IPF maps using the OIM software. “This can be explained by the increased heat input and high temperatures generated due to the excessively longer material flow path that affects the state of the material. “ as mentioned in the manuscript and reported by other researchers.
Comment #6
Section 3.3.1 must be proofread as some sentences are in concise
Response
The revised manuscript has been revised thoroughly and proofread. Thanks for the comment.
Comment#7
The authors presented the results obtained from texture analysis. However, the discussion is lagging. The role of plastic deformation on the preferred orientation should be discussed with their individual textural components Say, Cube, Bs, Cu, Dillamore and so on...
Response
Thanks for the critical comment. The texture section has been revised and improved with more discussion given and highlighted. The texture in FSW is mainly dominated by simple shear texture, with some components are shown in the manuscript. Also Table 3 for the ideal texture components has been added and the location of the ideal texture components are illustrated on all ODF sections. More explanation is given in the revised manuscript.
Comment #8
Hardness is decreasing but UTS is increasing. Explain.
Response
Thanks for the critical comment. The explanation for why hardness is decreasing is fairly covered in the revised manuscript and it is mainly because the starting material is work hardened and when experiencing heating during FSW, softening occurs and results in hardness reduction. In terms of tensile, the tensile properties after FSW is also lower than the BM because of the hardness reduction, but tensile not only depends on hardness but also on the grain size. Of course, we have grain refining, which slightly compensates for the reduction from hardness.
On addressing the above comments the paper can please be accepted for its publication

Reviewer 3 Report
This paper investigates the interesting problem of pin eccentricity at FSW. The material is chosen quite successfully because this alloy is thermally unstrengthenable. Nevertheless, I cannot agree that the work is advanced. There are a lot of studies like this. In the early days of FSW, there was a lot of research on the subject of eccentricity. Even in Mishra's encyclopedic review you can see these works. What's most interesting is that in the introduction, the authors do a good job of examining the material. But it is not clear from it why they did the research. All the conclusions they reached were known long ago. The work, of course, was carried out qualitatively. But it is not clear why it was carried out. I would like to hear the answers to these questions in the article. There are also more specific comments:
1. Very strange choice of eccentricity. 0,2 и 0,8. Analysis of Figure 14 suggests that an eccentricity of 0.4 would allow for better strength.
2. Figure 3: The quality of the drawings does not allow you to judge the quality of the surface. If you really want to show the difference, I suggest you take pictures at higher magnification.
3. Figure 5: You need to label the drawings with grain size and disorientations, and you need to label all drawings in the caption (Figure 5c is not listed right now).
4. Table 2: I would like to see the ranges of values.
5. Note to the whole article: the eccentric is there, but the mode is not compensated, because the friction rate is higher with a larger eccentric.
6. Figure 15: I suggest not connecting the point for Travel speed = 0, but making a separate line for the source material for clarity.
7. The pole figures are almost not described, although their images take up a lot of space.
Author Response
Response to comments of Reviewer 3
Dear Reviewer,
We do appreciate your positive feedback and the constructive comments you have provided that improved our manuscript significantly.
This paper investigates the interesting problem of pin eccentricity at FSW. The material is chosen quite successfully because this alloy is thermally unstrengthenable. Nevertheless, I cannot agree that the work is advanced. There are a lot of studies like this. In the early days of FSW, there was a lot of research on the subject of eccentricity. Even in Mishra's encyclopedic review you can see these works. What's most interesting is that in the introduction, the authors do a good job of examining the material. But it is not clear from it why they did the research. All the conclusions they reached were known long ago. The work, of course, was carried out qualitatively. But it is not clear why it was carried out. I would like to hear the answers to these questions in the article. There are also more specific comments:
Response
The aim of the article at the end of the introduction has been revised in a trial to answer the critical questions.
Comment #1
- Very strange choice of eccentricity. 0,2 и 0,8. Analysis of Figure 14 suggests that an eccentricity of 0.4 would allow for better strength.
Response
We do thank you for the suggestion and the analysis. When doing the work, the literature was investigated, and some recommendations been found that 0.4 eccentricity results in a reduction of tensile properties; thus the work planned to investigate a higher eccentricity to see of the reduction is continuing or the behavior can be different. But as per the reviewer's suggestion and analysis 0.4 eccentricity it worth investigation.
Comment #2
- Figure 3: The quality of the drawings does not allow you to judge the quality of the surface. If you really want to show the difference, I suggest you take pictures at higher magnification.
Response
Many thanks for the suggestions. The only difference that can be noted is the effect of traverse speed where at low speeds the surface is smooth with no clear circular features however, increasing the speed up to 500 mm/min resulted in clear spaced circular features due to the higher step per revolution. In order to clarify this and as per the reviewer's suggestion high mag images are presented for only the low speed (100mm/min) and the high speed (500mm/min) at the different tool eccentricities presented in Figure 3. Also, the corresponding transverse macrographs are presented in Figure 4
Comment #3
- Figure 5: You need to label the drawings with grain size and disorientations, and you need to label all drawings in the caption (Figure 5c is not listed right now).
Response
Many thanks for the comment and the important notice. This has been considered in the revised manuscript and all figure sections are labeled and defined in the caption and also described in the text.
Comment #4
- Table 2: I would like to see the ranges of values.
Response
These values are experimentally measured using two tensile samples from BM with no much difference between them thus the average is presented in this table. The range is given as plus-minus values as per the results.
Comment #5
- Note to the whole article: the eccentric is there, but the mode is not compensated, because the friction rate is higher with a larger eccentric.
Response
Many thanks for the comment. We agree that the friction rate is higher with a larger eccentrcity.
- Figure 15: I suggest not connecting the point for Travel speed = 0, but making a separate line for the source material for clarity.
Response
Thanks for the suggestion. It has been implemented.
Comment #7
- The pole figures are almost not described, although their images take up a lot of space.
Response
The section on crystallographic texture analysis has been substantially improved by adding table 3 for the ideal texture components and illustrating their positions in the ODF sections. More explanation and analysis is given.

Round 2
Reviewer 1 Report
The revised text can be accepted.
Reviewer 2 Report
The authors have taken efforts to address the comments and the responses given by them are satisfactory. Hence the manuscript can please be considered for its publication.
Reviewer 3 Report
Figure 5: typo "misorientation angles (d) distribution, and pole figures (d)"